# Comparing the Synergistic and Antagonistic Interactions of Ciprofloxacin and Levofloxacin Combined with Rifampin against Drug-Resistant *Staphylococcus aureus*: A Time–Kill Assay

**DOI:** 10.3390/antibiotics12040711

**Published:** 2023-04-06

**Authors:** Yu Ri Kang, Doo Ryeon Chung, Jae-Hoon Ko, Kyungmin Huh, Sun Young Cho, Cheol-In Kang, Kyong Ran Peck

**Affiliations:** 1Division of Infectious Diseases, Department of Medicine, Samsung Medical Center, Sungkyunkwan University School of Medicine, Seoul 06351, Republic of Korea; 2Asia Pacific Foundation for Infectious Diseases (APFID), Seoul 06351, Republic of Korea; 3Center for Infection Prevention and Control, Samsung Medical Center, Seoul 06351, Republic of Korea

**Keywords:** synergism, antagonism, fluoroquinolone, combination therapy, biofilm

## Abstract

Background: Treatment of device-related infections by drug-resistant *Staphylococcus aureus* can be challenging, and combination therapy has been proposed as a potential solution. We compared the effectiveness of levofloxacin–rifampin and ciprofloxacin–rifampin combinations in killing methicillin-resistant *S. aureus* (MRSA) using a time–kill assay. Methods: We randomly selected 15 vancomycin-susceptible *S. aureus* (VSSA) strains, 3 vancomycin-intermediate *S. aureus* (VISA) strains, and 12 heterogeneous VISA (hVISA) strains from the Asian Bacterial Bank. Time–kill experiments were performed in duplicate for each isolate. Viable bacterial counts were determined at 0 h, 4 h, 8 h, and 24 h for the ciprofloxacin– and levofloxacin–rifampin combinations at 1× MIC and 0.5× MIC. We compared synergistic and antagonistic interactions between the two combinations. Results: The viable bacterial count significantly decreased after 24 h of exposure to ciprofloxacin–rifampin and levofloxacin–rifampin combinations, with synergy observed more frequently in isolates exposed to ciprofloxacin–rifampin (43.3%) than levofloxacin–rifampin (20.0%) (*p* = 0.0082). The synergistic interactions of both combinations were more frequently observed in resistant strains with high MICs of ciprofloxacin (≥16 mg/L) and levofloxacin (≥8 mg/L). Levofloxacin tended to exhibit more frequent antagonistic interactions with rifampin than ciprofloxacin, although there was no statistical difference in antagonism between the two combinations. Conclusions: Our study demonstrated that ciprofloxacin exhibits superior synergistic activity against MRSA strains, including VISA/hVISA, when combined with rifampin compared with levofloxacin. High MICs of fluoroquinolones were found to predict synergism. Our results suggest that ciprofloxacin may be a more effective choice than levofloxacin for combination therapy with rifampin in the treatment of MRSA infections.

## 1. Introduction

*Staphylococcus aureus* is a leading pathogen responsible for a range of infections in both community and hospital settings, and its antibiotic resistance is on the rise [1]. Infections related to medical devices, such as prosthetic joints, are becoming increasingly common and are often challenging to treat even with bactericidal antibiotics [2,3]. Therefore, there is a critical need for new therapeutic strategies to combat these infections.

Combination therapy has emerged as a potential solution to tackle device-related infections caused by drug-resistant *S. aureus*, and many studies have examined the efficacy of combining different antibiotics. In particular, a combination of rifampin plus fluoroquinolone has been recommended for the treatment of staphylococcal prosthetic joint infection [4,5]. However, the benefit of this combination remains unclear. In vitro studies have yielded inconsistent results regarding the synergy of fluoroquinolones and rifampin, and some studies have even reported antagonistic effects [6]. Animal model studies have also produced mixed results regarding the efficacy of these antibiotic combinations in improving treatment outcomes [7,8]. Moreover, there is a lack of clinical evidence to support the combined use of rifampin in the treatment of methicillin-resistant *S. aureus* (MRSA) infections [9]. Most clinical trials have used beta-lactam or vancomycin in combination with rifampin [10,11,12], and there is currently no conclusive clinical evidence supporting the effectiveness of fluoroquinolone and rifampin combination therapy.

To address this gap in knowledge, our study aimed to investigate the potential synergistic or antagonistic effects of different fluoroquinolones in combination with rifampin and identify more effective treatment strategies for device-related infections caused by drug-resistant *S. aureus*. We compared the killing effects of the combination of levofloxacin and rifampin versus ciprofloxacin and rifampin against MRSA strains using a time–kill assay. 

## 2. Results

### 2.1. Antimicrobial Susceptibility

The MICs of rifampin for all MRSA strains ranged from 0.015 to 16 mg/L, with the VISA/hVISA strains exhibiting higher MIC_90_ values (16 mg/L) compared with VSSA strains (0.015 mg/L) (Table 1). Interestingly, the rate of rifampin resistance was found to be significantly higher in VISA/hVISA strains (40%) compared with VSSA strains (0%) (*p* = 0.0169).

In terms of ciprofloxacin and levofloxacin susceptibility, the MIC_50_ and MIC_90_ values did not show a significant difference between VSSA and VISA/hVISA groups. Similarly, the rates of resistance to ciprofloxacin and levofloxacin were not significantly different between the two groups (66.7% vs. 73.3%). It is worth noting that the MIC_50_ and MIC_90_ of levofloxacin were lower than those of ciprofloxacin, even though the resistance rates were equal between ciprofloxacin and levofloxacin (Table 1). 

### 2.2. Time–Kill Kinetics of Single Antibacterial Agents

In the time–kill kinetics experiments, single antibacterial agents alone, except for levofloxacin (1× MIC) at 8 h, did not significantly reduce the viable bacterial count or exhibit bactericidal activity at 4 h, 8 h, and 24 h compared with the initial inoculum (Figure 1 and Figure 2; Appendix A). The bactericidal activity of levofloxacin at 8 h was lost by 24 h, and the viable bacterial count at 24 h returned to the level of the initial inoculum, although the range was wide (Figure 1D and Appendix A). 

### 2.3. Synergistic Interaction

Compared with single antibacterial agents, the median viable bacterial count (log CFU/mL) significantly decreased at 24 h for ciprofloxacin– and levofloxacin–rifampin combinations, demonstrating synergistic interaction (Figure 1 and Figure 2; Appendix A). The ciprofloxacin–rifampin combination showed synergy at both 1× and 0.5× MIC concentrations, while the levofloxacin–rifampin combination showed synergy at the 1× MIC concentration. There was no statistically significant difference in bacterial count between the two combinations (Figure 3).

In 16 strains, the ciprofloxacin– or levofloxacin–rifampin combinations exhibited synergy as shown in Table 2. Notably, the highest rate of synergy was observed at 1× MIC concentration of rifampin combined with ciprofloxacin with a rate of 43.3%. This rate was significantly higher than that of rifampin combined with levofloxacin, which was only 20.0% (*p* = 0.0082) (Table 3). Only 1 of the 6 strains that showed a high MIC (16 mg/L) of rifampin exhibited synergy exclusively at 1× MIC concentration of rifampin combined with ciprofloxacin (Table 2). Among the 21 resistant strains with ciprofloxacin MICs of 4 mg/L or higher, 11 (52.4%) showed synergy at 1× MIC concentration of rifampin combined with ciprofloxacin. Interestingly, among the 13 strains that showed synergy with the ciprofloxacin–rifampin combination, only 2 were susceptible to ciprofloxacin (Table 2). We also found no difference in the proportions of strains showing synergy between VSSA and VISA/hVISA. The proportion of strains showing bactericidal activity was highest at 1× MIC concentration of rifampin combined with ciprofloxacin, and the rate was significantly higher than that of the levofloxacin–rifampin combination (36.7% vs. 20.0%, *p* = 0.0253) (Table 2).

### 2.4. Antagonistic Interaction

Antagonistic interactions were observed in a total of nine strains when tested with either ciprofloxacin–rifampin or levofloxacin–rifampin combinations (Table 2). Interestingly, antagonism appeared to be more common with the levofloxacin–rifampin combination than with the ciprofloxacin–rifampin combination (16.7% vs. 6.7% at 1× MIC; 16.7% vs. 10.0% at 0.5× MIC); however, this difference was not statistically significant (Table 3). No antagonism was observed in ciprofloxacin-resistant strains, and none of the VISA/hVISA strains showed antagonism at 1× MIC concentration of rifampin combined with ciprofloxacin.

### 2.5. Predictors of Synergistic and Antagonistic Interactions 

Further analysis using multiple logistic regression was conducted to identify predictors of synergistic and antagonistic interactions. The results showed that the combination of rifampin and ciprofloxacin had a synergistic effect 2.7 times more frequently in strains with a ciprofloxacin MIC of 16 mg/l or higher than in other strains (*p* = 0.0082). In contrast, a rifampin MIC of 16 mg/l or higher was negatively associated with synergism in the ciprofloxacin–rifampin combination (odds ratio, −2.559; *p* = 0.0478). In the levofloxacin–rifampin combination, a levofloxacin MIC of 8 mg/L or higher was found to predict synergism (odds ratio, 2.785; *p* = 0.0195). However, no predictors of antagonism for the two antibiotic combinations were found.

## 3. Discussion

In this study, we analyzed time–kill assay data to confirm the synergistic effect of ciprofloxacin–rifampin and levofloxacin–rifampin on the killing of drug-resistant *S. aureus* strains. Our findings suggest that ciprofloxacin exhibits a stronger synergistic effect when combined with rifampin than levofloxacin. Specifically, we observed a synergistic effect on the median viable bacterial count in both combinations of ciprofloxacin– and levofloxacin–rifampin at 1× MIC at 24 h. However, a synergistic effect at 0.5× MIC was found only in the ciprofloxacin–rifampin combination. Furthermore, a higher proportion of strains showed synergy with the ciprofloxacin–rifampin combination at 1× MIC than with the levofloxacin–rifampin combination.

Rifampin has been recommended as one of the therapeutic agents for the treatment of staphylococcal infections due to its potent bactericidal activity against *S. aureus* [4,5,13,14]. It has been shown to be highly effective in treating staphylococcal infections related to foreign bodies [15]. In addition, rifampin has been reported to retain activity against multidrug-resistant MRSA strains, making it a valuable option for the treatment of drug-resistant infections [16]. Notably, its efficacy against staphylococci in biofilm has been demonstrated in vitro, in animal models, and in patients with orthopedic device-related infections [14,17]. However, the use of rifampin as a single agent can lead to the development of resistance, highlighting the importance of combination therapy [13,18]. 

Fluoroquinolones have been considered the best antibiotics to use in combination with rifampin for the treatment of *S. aureus* infections [4,5,19,20,21]. However, studies have not shown consistent synergistic effects with the combination of fluoroquinolones and rifampin, and in some studies, antagonism has been reported [6]. Despite this, clinical studies on the combination of rifampin and fluoroquinolones have been conducted with ciprofloxacin, levofloxacin, and moxifloxacin as combination partners [22,23,24,25]. In particular, levofloxacin has often been regarded as an attractive combination partner in many observational studies [23,24], mainly because of its lower MIC against *S. aureus* strains and lower likelihood of developing resistance compared with ciprofloxacin [26,27]. 

The present study has addressed a critical knowledge gap concerning the optimal fluoroquinolone–rifampin combination for bacterial killing. The lack of research in this area underscores the importance of our study, which has revealed that the ciprofloxacin–rifampin combination exhibits a superior bacterial killing effect compared with the levofloxacin–rifampin combination. The slight difference in the frequency of antagonistic interaction between levofloxacin and rifampin, though not statistically significant, may provide a possible explanation for the observed difference in effect between the two combinations. This suggests that the use of levofloxacin in combination with rifampin may not be optimal, and that ciprofloxacin may be a better choice in this regard. It is also noteworthy that the rifampin–ciprofloxacin combination was not found to cause antagonism in ciprofloxacin-resistant MRSA strains. This observation highlights the potential benefits of using the rifampin–ciprofloxacin combination in the management of ciprofloxacin-resistant MRSA infections. Moreover, multiple logistic regression analyses identified high MIC values of ciprofloxacin and levofloxacin as predictors of synergistic interaction, which indicates that the effect of these antibiotic combinations may depend on the characteristics of the bacterial strain, such as the MIC values for each antibiotic.

Previous studies have also reported antagonism between rifampin and fluoroquinolone, particularly when combined with levofloxacin, which may lead to clinical hesitation in prescribing combination therapy [8,15,28]. Some animal studies have also shown antagonistic effects when either ciprofloxacin or levofloxacin is combined with rifampin [8,15,29]. The mechanism behind this antagonistic interaction between rifampin and fluoroquinolone is quite complex. It has been suggested that the inhibition of RNA synthesis by rifampin is responsible for abolishing the bactericidal killing activities of fluoroquinolone. This negative interaction is related to the activity against DNA supercoiling, which is a crucial step in the replication of bacterial DNA [15]. Consequently, the combination of rifampin and fluoroquinolone may reduce the effectiveness of fluoroquinolone in killing bacteria by compromising its bactericidal activity through decreased DNA supercoiling.

Although there are concerns about the potential antagonistic effects of combining rifampin and fluoroquinolone, previous studies have shown that this combination therapy can be beneficial in treating device-related infections. A study demonstrated that rifampin can antagonize the bactericidal effects of ciprofloxacin in staphylococci during exponential growth [28]. However, the study also showed that in non-growing cells, ciprofloxacin exhibited an additive bactericidal effect instead of antagonism [28]. This indicates that the effectiveness of this combination therapy may depend on the growth stage of the targeted bacteria. Therefore, caution should be exercised when administering rifampin and fluoroquinolone for therapeutic purposes in non-biofilm-related clinical situations, as the effects of this combination may vary depending on the specific circumstances of the infection.

This study has some limitations that need to be considered. Firstly, we only investigated the interaction of antibiotics at concentrations of 1× MIC and 0.5× MIC, which may not accurately predict interactions at higher concentrations. Secondly, our study only provides in vitro experimental results, and further clinical trials are necessary to draw conclusive recommendations for antibiotic combination therapy. 

Despite these limitations, our study is significant in its investigation of the interaction between rifampin and fluoroquinolone in combination for many drug-resistant *S. aureus* strains, including VISA/hVISA. These strains pose a significant challenge for treatment due to their drug resistance. Moreover, our study’s comparison of the interactions between ciprofloxacin and levofloxacin in combination with rifampin using various statistical analysis methods contributes to our understanding of the potential benefits and limitations of antibiotic combinations. In the co-administration of rifampin and fluoroquinolones, it should be noted that fluoroquinolones can cause aortic aneurysm in certain patients, as warned by the U.S. Food and Drug Administration (FDA) [30], and can cause persistent side effects in muscles and ligaments as warned by the European Medicines Agency (EMA) [31]. In addition, rifampin can reduce the concentration of other drugs through interaction when used in combination with certain drugs [32]. Therefore, this combination therapy, especially for critically ill patients, should be administered with caution, and good pharmacovigilance should be maintained.

In conclusion, our study provides evidence that ciprofloxacin exhibits superior synergistic activity against MRSA strains, including VISA/hVISA, when combined with rifampin compared with levofloxacin. Our study also highlights that the synergistic interactions between ciprofloxacin and rifampin, as well as levofloxacin and rifampin, are more frequently observed in resistant strains with high MICs of both fluoroquinolones. Interestingly, levofloxacin tended to exhibit more frequent antagonistic interactions with rifampin than ciprofloxacin. Overall, our results suggest that ciprofloxacin may be a more effective choice than levofloxacin for combination therapy with rifampin in the treatment of MRSA infections. However, further studies are needed to confirm these findings and determine the optimal strategy for combination therapy. These findings also underscore the importance of considering the potential synergistic or antagonistic interactions between antibiotics when selecting treatment regimens for MRSA infections.

## 4. Materials and Methods

### 4.1. Bacterial Isolates, Susceptibility Testing, and Genotyping

We tested a total of 30 MRSA strains. To ensure a comprehensive analysis, we specifically aimed to include strains with vancomycin non-susceptibility. To achieve this, we randomly selected 15 vancomycin-susceptible *S. aureus* (VSSA) strains, along with 3 vancomycin-intermediate *S. aureus* (VISA) strains and 12 heterogeneous VISA (hVISA) strains, from the MRSA bacterial collections stocked at the Asian Bacterial Bank (Asia Pacific Foundation for Infectious Diseases, Seoul, Republic of Korea). These collections included strains obtained from a previous nationwide bacteremia study in the Republic of Korea [33]. 

Species identification and initial susceptibility testing were performed using the VITEK^®^2 system (bioMérieux, Marcy-l’Étoile, France). We confirmed the initial susceptibility profile by broth microdilution according to Clinical and Laboratory Standards Institute (CLSI) guidelines [34]. *S. aureus* ATCC 29213 and *Enterococcus faecium* ATCC 29212 were utilized as control strains. We determined VISA and hVISA by the modified population analysis profile (PAP) method. The area under the curve (AUC) of the PAP graph was calculated and compared with that of reference strain Mu3 (ATCC 700698) [35]. We identified strains as VSSA, hVISA, or VISA based on the AUC_test_/AUC_Mu3_ ratio using these criteria: VSSA, <0.9; hVISA, 0.9 to 1.3; and VISA, >1.3. Mu50 (ATCC 700699) and Mu3 (ATCC 700698) were included as reference strains for VISA and hVISA, respectively.

Multilocus sequence typing (MLST) was conducted using polymerase chain reaction (PCR) amplification and sequencing of seven housekeeping genes (*arcC*, *aroE*, *glpF*, *gmk*, *pta*, *ypi*, and *yqiL*), as previously described [36]. The allelic profiles and sequence types (STs) were assigned according to the MLST web site (http://saureus.mlst.net/ (accessed on 3 April 2023)). Multiplex PCR was conducted for assignment of SCC*mec* types [37]. We performed *spa* typing as previously described [38] and using the Ridom SpaServer (http://spaserver.ridom.de (accessed on 3 April 2023)). 

### 4.2. Time–Kill Assay

To determine the killing effects of different antibiotic combinations against drug-resistant *S. aureus* strains, we performed time–kill assays. Each isolate was tested in duplicate using an inoculum of approximately 5 × 10^5^ CFU/mL in a final volume of 10 mL, as described previously [39]. Each isolate was tested against each antimicrobial agent alone and in combination at concentrations equal to 0.5× and 1× minimum inhibitory concentration (MIC). We specifically focused on combinations of rifampin with either ciprofloxacin or levofloxacin and evaluated their potential for synergy or antagonism. Samples were taken at 0 h, 4 h, 8 h, and 24 h after incubation with the indicated antimicrobials. To determine the number of viable bacterial colonies, dilutions were plated using an automatic spiral plater (Interscience, St. Nom, France), and the plates were incubated for 18 h to 24 h at 37 °C. We used a Scan 500 (Interscience, St. Nom, France) to count the bacterial colonies. An antibiotic-free growth control was included in each experiment. Time–kill curves were constructed by plotting mean colony count (log CFU/mL) versus time. Synergy was defined as a reduction of ≥2 log CFU/mL with the combination compared with the most active single agent and a reduction of ≥2 log CFU/mL below the initial inoculum at 24 h, as previously described [40]. Antagonism was defined as an increase of ≥2 log CFU/mL with the combination compared with the most active single agent at 24 h [40]. Bactericidal activity was defined as a reduction of ≥3 log CFU/mL compared with the initial inoculums [40].

### 4.3. Statistical Analysis

All statistical analyses were conducted using Stata version 11.2 (StataCorp, College Station, TX, USA) and R software (version 4.1.3). A *p* value less than 0.05 was considered statistically significant. To compare the differences in the antimicrobial resistance rates between VSSA and VISA/hVISA strains, we used both the Chi-square test and Fisher’s exact test, with Fisher’s exact test being utilized for any cell counts less than 5. To compare the proportion of bacterial strains showing synergy and antagonism between two antibiotic combinations, we used McNemar’s test, which is a statistical test used to compare paired proportions. The viable bacterial counts did not follow a normal distribution, as assessed by Shapiro–Wilk test. Therefore, non-parametric statistical tests were used to compare the medians of viable bacterial count between groups. We performed the Friedman test, followed by the Wilcoxon signed-rank test with Bonferroni adjustment for multiple groups. Multiple logistic regression analysis was performed to identify predictors for synergistic actions of ciprofloxacin– and levofloxacin–rifampin combinations. This analysis included variables such as year of isolation, vancomycin susceptibility, sequence type, SCC*mec* type, *spa* type, and levels of resistance to rifampin, ciprofloxacin, and levofloxacin.

## Figures and Tables

**Figure 1 antibiotics-12-00711-f001:**
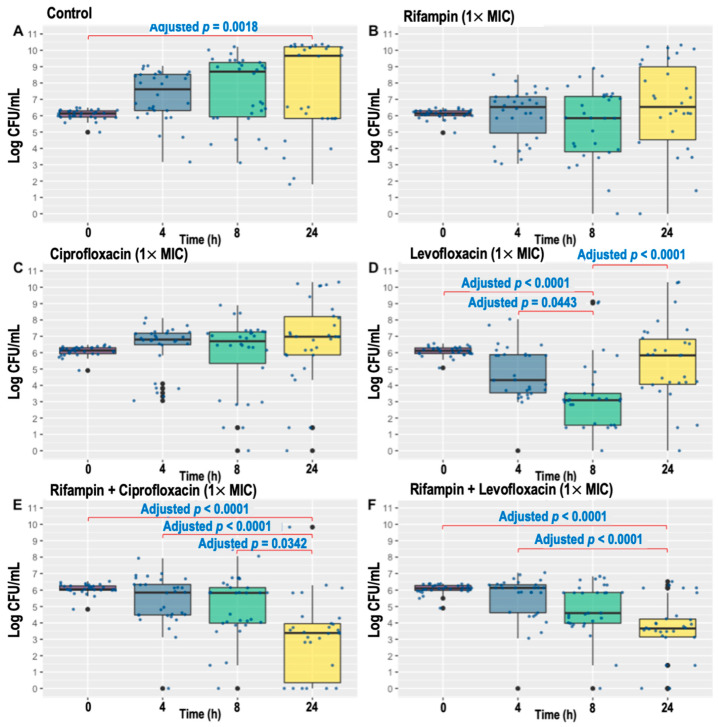
Changes in viable bacterial counts over 24 h after exposure to antibacterial agents at a concentration of 1× MIC (minimum inhibitory concentration). (**A**) Changes in viable bacterial counts after exposure to control. (**B**) Changes in viable bacterial counts after exposure to rifampin. (**C**) Changes in viable bacterial counts after exposure to ciprofloxacin. (**D**) Changes in viable bacterial counts after exposure to levofloxacin. (**E**) Changes in viable bacterial counts after exposure to rifampin and ciprofloxacin. (**F**) Changes in viable bacterial counts after exposure to rifampin and levofloxacin.

**Figure 2 antibiotics-12-00711-f002:**
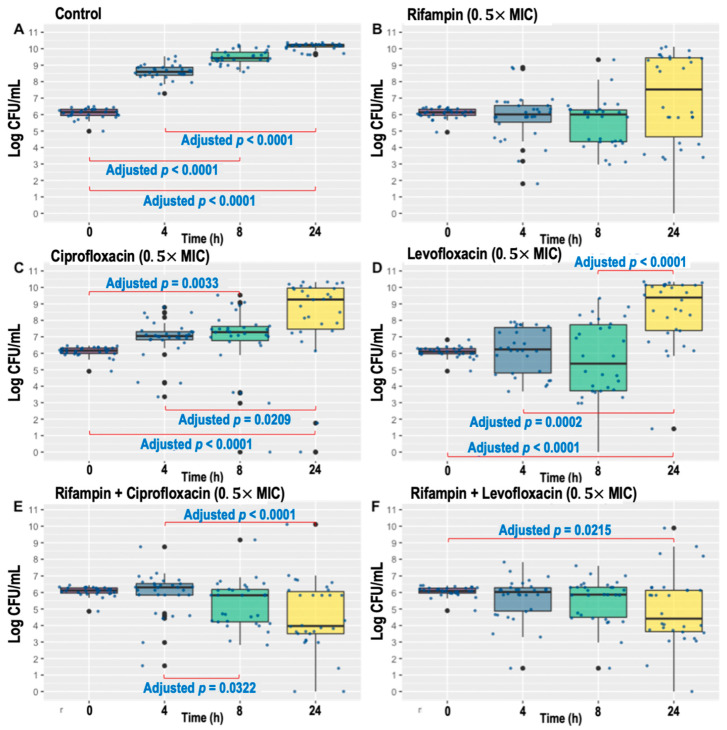
Changes in viable bacterial counts over 24 h after exposure to antibacterial agents at a concentration of 0.5× MIC (minimum inhibitory concentration). (**A**) Changes in viable bacterial counts after exposure to control. (**B**) Changes in viable bacterial counts after exposure to rifampin. (**C**) Changes in viable bacterial counts after exposure to ciprofloxacin. (**D**) Changes in viable bacterial counts after exposure to levofloxacin. (**E**) Changes in viable bacterial counts after exposure to rifampin and ciprofloxacin. (**F**) Changes in viable bacterial counts after exposure to rifampin and levofloxacin.

**Figure 3 antibiotics-12-00711-f003:**
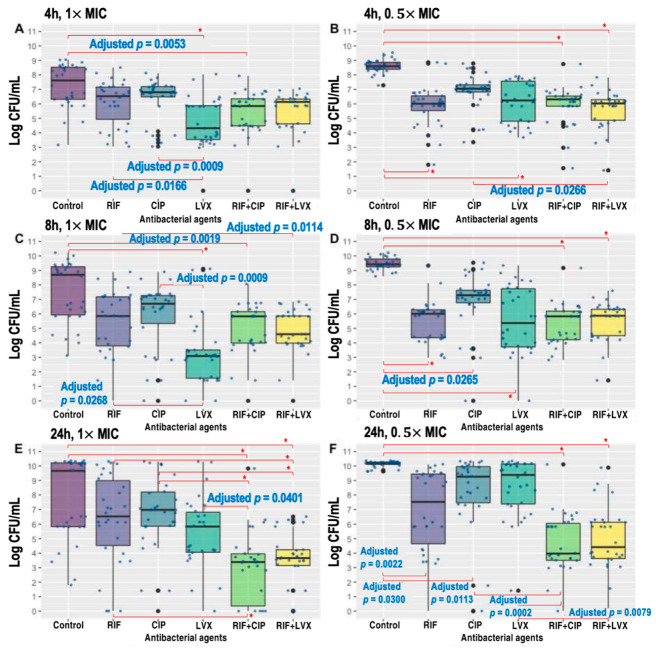
Comparison of viable bacterial counts after exposure to different antibacterial agents for a specific time period. * Adjusted *p* < 0.0001. (**A**) Changes in viable bacterial counts following a 4-h exposure to antibacterial agents at a concentration equal to 1× minimum inhibitory concentration (MIC). (**B**) Changes in viable bacterial counts following a 4-h exposure to antibacterial agents at a concentration equal to 0.5× minimum inhibitory concentration (MIC). (**C**) Changes in viable bacterial counts following a 8-h exposure to antibacterial agents at a concentration equal to 1× minimum inhibitory concentration (MIC). (**D**) Changes in viable bacterial counts following a 8-h exposure to antibacterial agents at a concentration equal to 0.5× minimum inhibitory concentration (MIC). (**E**) Changes in viable bacterial counts following a 24-h exposure to antibacterial agents at a concentration equal to 1× minimum inhibitory concentration (MIC). (**F**) Changes in viable bacterial counts following a 24-h exposure to antibacterial agents at a concentration equal to 0.5× minimum inhibitory concentration (MIC).

**Table 1 antibiotics-12-00711-t001:** MIC distribution of antimicrobials against methicillin-resistant *Staphylococcus aureus* strains.

	Rifampin			Ciprofloxacin			Levofloxacin		
	MIC_50_ (mg/L)	MIC_90_ (mg/L)	Resistance Rate (%)	MIC_50_ (mg/L)	MIC_90_ (mg/L)	Resistance Rate (%)	MIC_50_ (mg/L)	MIC_90_ (mg/L)	Resistance Rate (%)
VSSA (n = 15)	0.015	0.015	0	32	>64	66.7	8	32	66.7
VISA/hVISA (n = 15)	0.015	16	40.0	16	>64	73.3	8	32	73.3

MIC, minimum inhibitory concentration; VSSA, vancomycin-susceptible *S. aureus*; VISA, vancomycin-intermediate *S. aureus*; hVISA, heterogeneous VISA.

**Table 2 antibiotics-12-00711-t002:** Microbiological and molecular characteristics of methicillin-resistant *Staphylococcus aureus* strains and comparison of rifampin interactions with ciprofloxacin and levofloxacin.

Isolate	Phenotype	Year	ST	SCC*mec* Type	*spa* Type	MIC (mg/L)			Interaction				Bactericidal Activity			
									0.5× MIC		1× MIC		0.5× MIC		1× MIC	
						R	C	L	R + C	R + L	R + C	R + L	R + C	R + L	R + C	R + L
1	VSSA	2006	239	III	t037	0.015	32	8	I	I	I	I	N	N	B	N
2	VSSA	2006	72	IVA	t324	0.015	0.5	0.25	I	I	A	A	N	N	N	N
3	VSSA	2006	239	IIIA	t037	0.015	>64	32	I	A	Syn	I	B	N	B	B
4	VSSA	2007	72	IVA	t148	0.015	0.5	0.25	A	A	A	A	N	N	N	N
5	VSSA	2007	239	III	t037	0.015	64	16	Syn	I	Syn	I	N	N	N	N
6	VSSA	2007	5	II	t002	0.015	>64	>32	I	I	Syn	Syn	N	N	N	N
7	VSSA	2007	5	II	t002	0.015	>64	>32	I	I	Syn	Syn	N	N	N	N
8	VSSA	2007	239	III	t037	0.015	64	32	Syn	Syn	Syn	Syn	N	N	B	N
9	VSSA	2012	5	II	t9353	0.015	>64	32	I	I	Syn	I	N	N	N	N
10	VSSA	2012	72	IVA	t148	0.015	0.5	0.25	I	I	I	I	N	N	N	N
11	VSSA	2012	5	II	t2460	0.015	32	8	I	A	I	I	N	N	N	N
12	VSSA	2012	72	IVA	t324	0.015	0.25	0.25	I	I	I	I	N	N	N	N
13	VSSA	2012	5	II	t9353	0.015	64	32	Syn	Syn	I	I	B	N	N	N
14	VSSA	2013	72	IVA	t148	0.015	0.25	0.25	I	I	Syn	I	N	B	N	N
15	VSSA	2013	239	III	t138	1	4	4	I	I	I	I	N	N	N	N
16	VISA	2008	239	IIIA	t037	16	16	8	I	I	Syn	I	N	N	B	B
17	VISA	2009	5	II	t2460	16	64	16	A	I	I	A	N	N	B	N
18	VISA	2011	72	IVA	t324	16	1	0.25	A	I	I	I	B	B	B	B
19	hVISA	2006	72	IVA	t324	0.015	0.25	0.5	I	I	I	I	N	N	N	N
20	hVISA	2006	5	II	t002	0.015	16	8	Syn	Syn	I	I	N	N	N	N
21	hVISA	2006	5	II	t2460	0.015	16	8	Syn	I	Syn	Syn	N	N	N	N
22	hVISA	2006	239	III	t037	0.015	16	8	Syn	Syn	Syn	Syn	N	N	N	N
23	hVISA	2007	5	II	t2460	16	>64	16	I	I	I	I	N	N	B	N
24	hVISA	2007	239	III	t037	0.015	32	8	Syn	Syn	Syn	A	B	B	B	N
25	hVISA	2007	239	III	t037	16	8	8	I	I	I	I	N	N	B	B
26	hVISA	2008	5	II	t601	0.015	>64	32	I	Syn	I	I	N	N	N	N
27	hVISA	2010	72	IVA	t148	0.015	0.5	0.5	Syn	Syn	Syn	Syn	B	B	B	B
28	hVISA	2011	72	IVA	t324	0.015	0.25	0.25	I	A	I	A	N	N	N	N
29	hVISA	2011	5	II	t9353	0.015	32	8	I	I	Syn	I	N	N	N	N
30	hVISA	2013	5	II	t9353	16	>64	32	I	A	I	I	N	N	B	B

VSSA, vancomycin-susceptible *S. aureus*; VISA, vancomycin-intermediate *S. aureus*; hVISA, heterogeneous VISA; ST, sequence type; SCC, staphylococcal cassette chromosome; MIC, minimum inhibitory concentration; R, rifampin; C, ciprofloxacin; L, levofloxacin; I, indifference; A, antagonism; Syn, synergy; B, bactericidal; N, non-bactericidal.

**Table 3 antibiotics-12-00711-t003:** Comparison of synergistic and antagonistic interactions of ciprofloxacin and levofloxacin combined with rifampin against methicillin-resistant *Staphylococcus aureus* strains.

Interaction	Number of Bacterial Strains (%)					
	1× MIC			0.5× MIC		
	Rifampin + Ciprofloxacin	Rifampin + Levofloxacin	*p* Value	Rifampin + Ciprofloxacin	Rifampin + Levofloxacin	*p* Value
Synergy	12 (43.3%)	6 (20.0%)	0.0082	8 (26.7%)	7 (23.3%)	0.5637
Antagonism	2 (6.7%)	5 (16.7%)	0.0833	3 (10.0%)	5 (16.7%)	0.4142

MIC, minimum inhibitory concentration.

## Data Availability

The data presented in this study are available on request from the corresponding author.

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
