# Peer review of "Comparing the Synergistic and Antagonistic Interactions of Ciprofloxacin and Levofloxacin Combined with Rifampin against Drug-Resistant Staphylococcus aureus: A Time–Kill Assay"

_antibiotics, 2023, doi:10.3390/antibiotics12040711_

Round 1

Reviewer 1 Report

This work provides a new and important combinations to effectively kill the methicillin-resistant S. aureus using a time-kill assay in vitro, which would ultimately result in efficient treatment of the infections by the drug-resistant S. aureus after being verified by animal experiments and clinical trials.

Author Response

(x) English language and style are fine/minor spell check required

-> As you pointed out, we carefully reviewed the manuscript and further improved the English sentences, as well as conducted a spell check.

Reviewer 2 Report

The paper is very well written and addresses the concern about the antimicrobial resistance very well. However, there are few comments which needs to be addressed. 

In line no 129, authors have mentioned that for the statistical analysis, they have used chi square/Fisher test; however it is still not clear which among the two tests was used. Please clarify the same.

In all the figures, P value needs to be highlighted.

In the references, more recent communication of 2022 and 2023, if any needs to be added

Author Response

The paper is very well written and addresses the concern about the antimicrobial resistance very well. However, there are few comments which needs to be addressed. 

1) In line no 129, authors have mentioned that for the statistical analysis, they have used chi square/Fisher test; however it is still not clear which among the two tests was used. Please clarify the same.

-> Based on the comment, we revised the sentence in the statistical analysis part of the Method section as follows:

“To compare the differences in the antimicrobial resistance rates between VSSA and VISA/hVISA strains, we used both the Chi-square test and Fisher’s exact test, with Fisher’s exact test being utilized for any cell counts less than 5.”

2) In all the figures, P value needs to be highlighted.

->  We revised all three figures to highlight P values within the figures for better visibility.

3) In the references, more recent communication of 2022 and 2023, if any needs to be added.

->  As you pointed out, we carefully reviewed all the references and replaced references 2 and 5 with sources published in 2023 and 2022, respectively.

References

  1. Tuon FF, Suss PH, Telles JP, Dantas LR, Borges NH, Ribeiro VST. 2023. Antimicrobial treatment of Staphylococcus aureus Antibiotics 12:87.

  1. Cortes-Penfield NW, Hewlett AL, Kalil AC. 2022. Adjunctive rifampin following debridement and implant retention for staphylococcal prosthetic joint infection: Is it effective if not combined with a fluoroquinolone? Open Forum Infect Dis 9:ofac582.

Reviewer 3 Report

In the manuscript entitled "Comparing the synergistic and antagonistic interactions of ciprofloxacin and levofloxacin combined with rifampin against drug-resistant Staphylococcus aureus: a time-kill assay" authors compared the effectiveness of levofloxacin-rifampin and ciprofloxacin-rifampin combinations in killing methicillin-resistant Staphylococcus aureus (15 vancomycin-susceptible S. aureus strains, 3 vancomycin-intermediate S. aureus strains, and 12 heterogeneous vancomycin-intermediate S. aureus strains) using a time-kill assay. The results of this study suggest that ciprofloxacin may be a more effective choice than levofloxacin for combination therapy with rifampin in the treatment of MRSA infections. The manuscript can be accepted after some minor changes.

1. Include more recent literature. 

2. Results should be presented as mean +/- SD/SE.

3. Discuss rationale of the study and rationale behind selection of strains. 

Author Response

  1. Include more recent literature. 

-> As you pointed out, we carefully reviewed all the references and replaced references 2 and 5 with sources published in 2023 and 2022, respectively.

References

  1. Tuon FF, Suss PH, Telles JP, Dantas LR, Borges NH, Ribeiro VST. 2023. Antimicrobial treatment of Staphylococcus aureus Antibiotics 12:87.

  1. Cortes-Penfield NW, Hewlett AL, Kalil AC. 2022. Adjunctive rifampin following debridement and implant retention for staphylococcal prosthetic joint infection: Is it effective if not combined with a fluoroquinolone? Open Forum Infect Dis 9:ofac582.

  1. Results should be presented as mean +/- SD/SE.

-> Thank you for your helpful comments on our methods and presentation of results in the study, which will improve the clarity of our manuscript.  As viable bacterial counts did not show a normal distribution, we presented the median and range. We also compared the median values rather than the means when comparing between groups. We have added this information to the statistical analysis part of the Method section. We also would like to note that Supplementary Fig. 1, which was originally included in the manuscript, displays mean (CFU/ml) with standard deviation on the graph.

“The viable bacterial counts did not follow a normal distribution, as assessed by Shapiro-Wilk test. Therefore, non-parametric statistical tests were used to compare the medians of viable bacterial count between groups.”

  1. Discuss rationale of the study and rationale behind selection of strains. 

-> Thank you for your helpful comments. We realized that the part you mentioned was somewhat weak in the original manuscript. Accordingly, we revised the introduction section as follows, and made some modifications to the bacterial strain selection part in the method section in response to your suggestion.

  1. Introduction

Staphylococcus aureus is a leading pathogen responsible for a range of infections in both community and hospital settings, and its antibiotic resistance is on the rise [1]. Infections related to medical devices, such as prosthetic joints, are becoming increasingly common and are often challenging to treat even with bactericidal antibiotics [2,3]. Therefore, there is a critical need for new therapeutic strategies to combat these infections.

Combination therapy has emerged as a potential solution to tackle device-related infections caused by drug-resistant S. aureus, and many studies have examined the efficacy of combining different antibiotics. In particular, a combination of rifampin plus fluoroquinolone has been recommended for the treatment of staphylococcal prosthetic joint infection [4,5]. However, the benefit of this combination remains unclear. In vitro studies have yielded inconsistent results regarding the synergy of fluoroquinolones and rifampin, and some studies have even reported antagonistic effects [6]. Animal model studies have also produced mixed results regarding the efficacy of these antibiotic combinations in improving treatment outcomes [7,8]. Moreover, there is a lack of clinical evidence to support the combined use of rifampin in the treatment of methicillin-resistant S. aureus (MRSA) infections [9]. Most clinical trials have used beta-lactam or vancomycin in combination with rifampin [10-12], and there is currently no conclusive clinical evidence supporting the effectiveness of fluoroquinolone and rifampin combination therapy.

To address this gap in knowledge, our study aimed to investigate the potential synergistic or antagonistic effects of different fluoroquinolones in combination with rifampin and identify more effective treatment strategies for device-related infections caused by drug-resistant S. aureus. We compared the killing effects of the combination of levofloxacin and rifampin versus ciprofloxacin and rifampin against MRSA strains using a time-kill assay.

  1. Materials and Methods

2.1. Bacterial isolates, susceptibility testing and genotyping

We tested a total of 30 MRSA strains. To ensure a comprehensive analysis, we specifically aimed to include strains with vancomycin non-susceptibility. To achieve this, we randomly selected 15 vancomycin-susceptible S. aureus (VSSA) strains, along with three vancomycin-intermediate S. aureus (VISA) strains and 12 heterogeneous VISA (hVISA) strains, from the MRSA bacterial collections stocked at the Asian Bacterial Bank (Asia Pacific Foundation for Infectious Diseases, Seoul, South Korea). These collections included strains obtained from a previous nationwide bacteremia study in South Korea [13].